# Activation of Gingival Fibroblasts by Bacterial Cyclic Dinucleotides and Lipopolysaccharide

**DOI:** 10.3390/pathogens9100792

**Published:** 2020-09-26

**Authors:** Samira Elmanfi, Herman O. Sintim, Jie Zhou, Mervi Gürsoy, Eija Könönen, Ulvi K. Gürsoy

**Affiliations:** 1Department of Periodontology, Institute of Dentistry, University of Turku, 20520 Turku, Finland; samira.h.elmanfi@utu.fi (S.E.); mervi.gursoy@utu.fi (M.G.); eija.kononen@utu.fi (E.K.); 2Department of Chemistry and Purdue Institute for Drug Discovery and Purdue Institute of Inflammation, Immunology and Infectious Disease, Purdue University, West Lafayette, IN 47907, USA; cindyowen3447@gmail.com; 3Oral Health Care, Welfare Division, City of Turku, 20520 Turku, Finland

**Keywords:** lipopolysaccharide, inflammation, interleukin, matrix metalloproteinases, human gingival fibroblasts

## Abstract

Human gingival fibroblasts (HGFs) recognize microbe-associated molecular patterns (MAMPs) and respond with inflammatory proteins. Simultaneous impacts of bacterial cyclic di-guanosine monophosphate (c-di-GMP), cyclic di-adenosine monophosphate (c-di-AMP), and lipopolysaccharide (LPS) on gingival keratinocytes have been previously demonstrated, but the effects of these MAMPs on other periodontal cell types, such as gingival fibroblasts, remain to be clarified. The present aim was to examine the independent and combined effects of these cyclic dinucleotides and LPS on interleukin (IL) and matrix metalloproteinase (MMP) response of HGFs. The cells were incubated with c-di-GMP and c-di-AMP, either in the presence or absence of *Porphyromonas gingivalis* LPS, for 2 h and 24 h. The levels of IL-8, -10, and -34, and MMP-1, -2, and -3 secreted were measured by the Luminex technique. LPS alone or together with cyclic dinucleotides elevated IL-8 levels. IL-10 levels were significantly increased in the presence of c-di-GMP and LPS after 2 h but disappeared after 24 h of incubation. Concurrent treatment of c-di-AMP and LPS elevated MMP-1 levels, whereas c-di-GMP with LPS suppressed MMP-2 levels but increased MMP-3 levels. To conclude, we produce evidence that cyclic dinucleotides interact with LPS-mediated early response of gingival fibroblasts, while late cellular response is mainly regulated by LPS.

## 1. Introduction

Human gingival fibroblasts (HGFs) contribute to the maintenance of the gingival tissue homeostasis by taking part in repair and regeneration. These cells can also modulate immune response against oral pathogenic bacteria by sensing the shifts in the microbe-associated molecular patterns (MAMPs) [1,2,3]. MAMPs are carbohydrates, lipoproteins, or nucleic acids expressed as part of the bacterial life cycle [4]. A well-known example of oral MAMPs is the lipopolysaccharide (LPS) of *Porphyromonas gingivalis* (*P. gingivalis*), which acts as a potent stimulant to the host immune system [5,6]. Recognition of MAMPs via pathogen recognition receptors (PRRs) leads to the secretion of a wide range of inflammatory mediators, including cytokines (interleukins, ILs), enzymes (matrix metalloproteinases, MMPs), and antimicrobial peptides [2]. PRRs include four different classes: toll-like receptors (TLRs), c-type lectin receptors (CLRs), retinoic acid-inducible gene (RIG)-I-like receptors (RLRs), and NOD-like receptors (NLRs) [7]. TLRs exhibit specificity for ligand recognition, for instance, TLR4 can sense especially *P. gingivalis* LPS [8,9,10]. In response to *P. gingivalis* LPS, HGFs produce and secrete cytokines (IL-6, IL-8, IL-10, IL-34) and enzymes (MMP-1, MMP-2, MMP-3), which eventually participate in the breakdown of periodontal tissues [11,12].

Bacterial cyclic di-guanosine monophosphate (c-di-GMP) and cyclic di-adenosine monophosphate (c-di-AMP) are important secondary signaling molecules [13]. C-di-GMP is produced by Gram-negative bacteria and regulates a wide range of central bacterial processes, including transition from biofilm to motility [14], while c-di-AMP is produced by Gram-positive bacteria and has regulatory roles in cell wall homeostasis, DNA repair, and biofilm formation [15,16]. In addition to their roles in bacterial intracellular signaling pathways, cyclic dinucleotides stimulate innate immune response in mammalian cells via the Stimulator of Interferon Genes (STING) pathway [17,18]. Upon activation, STING stimulates the production of TANK-binding kinase 1 (TBK1) and interferon regulatory factors (IRF3). This, in turn, stimulates the expression of proinflammatory cytokines via induction of the transcription of type I interferons and the nuclear factor-κB (NF-Κb) [18]. Various effects of bacterial cyclic dinucleotides on cellular response of leukocytes and macrophages have been demonstrated [19,20,21].

In the oral cavity, the action of multiple MAMPs on host cells is continuous and simultaneous. We recently demonstrated that in gingival keratinocytes, c-di-AMP neutralizes the LPS-inhibited IL-8 expression [22]. Despite the vital role of fibroblasts in the amplification and perpetuation of the inflammatory response, to our knowledge, the synchronous action of multiple MAMPs on gingival fibroblasts has not been studied so far. According to our hypothesis, stimulation of gingival fibroblast inflammatory response by bacterial cyclic dinucleotides and LPS induces different outcomes when these MAMPs are used alone or together. In the present study, we examined if c-di-GMP and c-di-AMP interact with *P. gingivalis* LPS-regulated cellular response of gingival fibroblasts.

## 2. Results

Levels of IL-8 increased significantly when HGFs were cultured with either LPS alone (*p* < 0.001) or with c-di-GMP for 2 h (*p* < 0.001 at concentrations of 100 µM, 10 µM, and 1 µM) and for 24 h (*p* < 0.001 at concentrations of 100 µM and 10 µM, and *p* = 0.002 at 1 µM concentration). Incubation of HGFs with LPS together with c-di-AMP (*p* < 0.001 at 100 µM) for 2h, or with LPS alone (*p* < 0.001) for 24 h, or with any of the tested c-di-AMP concentrations together with LPS for 24 h elevated IL-8 levels markedly (Figure 1A,B).

IL-10 concentrations were significantly increased when HGFs were exposed to c-di-GMP in the presence of LPS for 2 h (at 10 µM and 1 µM *p* < 0.001). Instead, after 24 h of incubation, IL-10 concentrations elevated significantly only when HGFs were cultured with c-di-GMP alone (*p* = 0.001 at 100 µM) (Figure 2A,B). No marked differences were observed in concentrations of IL-34 when HGFs were incubated with any of the tested MAMPs for 2 h and 24 h (Figure 3A,B). 

MMP-1 concentrations did not change after HGFs were incubated with LPS or with any of the cyclic dinucleotides for 2 h. Incubation of HGFs with LPS and c-di-AMP (at 100 µM and 10 µM *p* = 0.01) for 24 h elevated the concentrations of MMP-1 significantly (Figure 4A,B).

Incubation of HGFs with 10 µM of c-di-GMP and LPS for 2 h (*p* = 0.003) stimulated MMP-2 secretion, while the incubation for longer period (24 h) with 1 µM of c-di-AMP (*p* = 0.04), 1 µM c-di-GMP (*p* = 0.004), and c-di-GMP (at 100 µM and 10 µM *p* = 0.01, and at 1 µM *p* < 0.001) with LPS decreased the concentrations of MMP-2 (Figure 5A,B). MMP-3 levels enhanced significantly after incubation of HGFs with c-di-GMP and LPS for 2 h (at 10 µM and 1 µM *p* < 0.001) and for 24 h (at 100 µM *p* = 0.009, at 10 µM *p* = 0.03) (Figure 6A,B). 

## 3. Discussion

Here, we demonstrated the difference between early and late cytokine and MMP response in gingival fibroblasts against multiple MAMPs. Cyclic dinucleotides interacted with LPS-mediated early cellular response (2 h), while late response (24 h) was mainly regulated by LPS. To our knowledge, no previous study has demonstrated cyclic dinucleotide-regulated cellular response of human gingival fibroblasts. 

Expression of second messenger molecules by bacteria to adapt environmental changes has been investigated [23], but the regulatory effect of these bacterial molecules on cellular response of eukaryotic cells was poorly characterized. Our group recently showed that the STING pathway in macrophages is related to their phagocytic behavior, indicating that bacterial cyclic dinucleotides may take part in direct bacterial elimination by host cells [20]. Although the TLR and STING pathways are crucial features in mammalian innate immunity, they may also compete with each other to induce or suppress host defense. There is evidence to suggest that these pathways play agonistic or opposite roles in host defense, depending on the type of infection and host-model used [24,25]. The main strength of the present study is that both combined and independent actions of multiple MAMPs on gingival fibroblasts were examined. Moreover, cellular response of HGF were evaluated at two different time points. As previously suggested, immediate response of fibroblasts against lipid mediators and prostaglandins start within minutes [26]. Peptide messengers mediate the early cellular response, which develop within hours. Finally, growth factors and extracellular matrix proteins regulate late responses that develop within hours to days [26]. Similar time-dependent shifts in cellular response were also demonstrated when HGFs were incubated with *P. gingivalis* LPS [27,28,29]. For that reason, we applied two time points (2 h for early and 24 h for late cellular response) to follow the interactions of cyclic dinucleotides with LPS mediated HGF response.

Studies on skin fibroblasts indicated that these cells have higher TLR expression profiles than skin keratinocytes [30]. Thus, there is an obvious need for investigating independent and combined effects of multiple MAMPs on gingival fibroblasts at different time points. The selection of cytokines and enzymes to be tested was based on their roles in immune response; IL-8, a chemoattractant cytokine, induces the infiltration of neutrophils and the release of other inflammatory cytokines such as IL-6 in periodontal tissues [31], IL-34 contributes to osteoclastogenesis in periodontitis [32], and IL-10, an anti-inflammatory and immunosuppressive cytokine, protects against tissue destruction via inhibiting MMPs and receptor activator for nuclear factor-kB (RANK) systems [33]. 

Limitations of the present study are related to its in vitro design and the use of a monolayer cell culture. Furthermore, fibroblasts were stimulated with MAMPs instead of live bacteria, which may be seen as a limitation, since in in vivo conditions, live bacteria activate and suppress various eukaryotic pathways simultaneously. Additionally, the use of commercially obtained LPS is a limiting factor, as these preparations may contain trace amounts of lipoproteins that can activate TLR-2 response. However, live bacteria also harbor other immune stimulants (such as peptidoglycan fragments and DNA) [34,35]. Therefore, it had been impossible to tease out the combined or individual roles of LPS and cyclic dinucleotides if live bacteria were used. In that regard, the in vitro design is not necessarily a limitation, instead it could be the best approach available to study individual or combined effects of two MAMPs.

LPS is the endotoxin component of Gram-negative bacteria and one of the main pathogenic activators of the host immune system. LPS induces an immune response by activating the mitogen-activated protein kinases (MAPKs) to produce proinflammatory cytokines [36,37]. According to our results, a meaningful increase in IL-8 expression occurs after incubation of HGFs with *P. gingivalis* LPS. It was previously demonstrated that *P. gingivalis* LPS stimulates gingival fibroblasts to express mRNA of IL-8 [38] and enhances IL-8 production without any changes in IL-10 levels [39], which was confirmed in our study. *P. gingivalis* can downregulate fibroblasts’ cellular response by degrading a wide range of inflammatory mediators [40]. In response to *P gingivalis* LPS, MMPs are synthesized and produced from HGFs, which eventually participate in breakdown of periodontal tissues [41]. In the present study, no marked changes were observed at MMP levels, when HGFs were treated with LPS alone. Increase in the distinct function and mRNA expression of MMPs in response to *P. gingivalis* LPS is dependent on the applied LPS dose and incubation time [42]. For example, elevated MMP-2 mRNA expression was observed after 72 h of incubation of HGFs with 30 ng/mL LPS, whereas MMP-1 mRNA expression was elevated significantly when the cells were incubated with 3000 ng/mL LPS for 8 days [42]. Thus, the difference between the current and previous study outcomes may be due to the different exposure time and LPS concentrations used. 

Gingival fibroblasts produce both proinflammatory (IL-8 and IL-34) and anti-inflammatory (IL-10) cytokines. In the present study, the simultaneous application of c-di-AMP and LPS stimulated IL-8 secretion after 2 h of incubation only at the highest c-di-AMP concentration. When the cells were incubated for 24 h, this effect was seen at all tested c-di-AMP concentrations. No changes were observed when the cells were incubated with c-di-AMP alone. These findings indicate that IL-8 secretion is mainly regulated by LPS, while c-di-AMP has no direct effect on it. A stimulating activity of c-di-AMP in host immune response can be recognized by an endoplasmic reticulum membrane adaptor that activates nuclear factor kappa-light-chain-enhancer of activated B cells (NF-κB) and enhances the production of proinflammatory cytokines [43,44]. In the present study, incubation of HGFs with c-di-GMP and LPS resulted in a significant increase in IL-8 (after 2 h and 24 h of incubation) and IL-10 (after 2 h of incubation) levels. Interestingly, c-di-GMP alone significantly elevated IL-10 levels after 24 h of incubation. As an anti-inflammatory and immunosuppressive cytokine, IL-10 protects periodontal tissues against destruction [33]. C-di-GMP has an immunostimulatory action on innate immune cells, such as monocytes, macrophages, and granulocytes, and it also induces the maturation of dendritic cells to produce various cytokines and chemokines [21]. Expression of IL-8, but not that of IL-10, in dendritic cells can also be triggered by c-di-GMP [21]. We have previously shown that c-di-AMP is able to neutralize the effects of LPS on IL-8 response in human gingival keratinocytes, while no changes were observed when these cells were incubated with c-di-GMP [22]. To our knowledge, IL-10 activating mechanisms of c-di-GMP on HGFs have not been demonstrated. Its contribution to healthy homeostasis and disease pathogenesis requires further investigation.

MMPs contribute to tissue remodeling and turnover of the periodontium [41,45]. MMP-1, -2, and -3 are known to be involved in connective tissue breakdown and to modulate expression of cytokines [45,46]. According to our results, incubation of HGFs with LPS or c-di-AMP alone for 24 h did not elevate MMP-1 levels, whereas the simultaneous application of c-di-AMP and LPS produced a significant increase. This indicates that the cellular response of fibroblasts stimulated by multiple actions of MAMPs is different from any individual MAMP. Based on our results, MMP-2 levels are stimulated by a combined application of c-di GMP and LPS for 2 h, whereas a longer incubation time leads to the suppression of this enzyme. The relation between MMPs and inflammatory cytokines is reciprocal; these proteins can suppress or activate each other’s expressions. Therefore, the suppression of MMP-2 secretion may be directly related to MAMP regulation or to the upregulation of proinflammatory cytokine expression [47]. Moreover, MMPs take part in post-transcriptional processing of inflammatory cytokines and chemokines, which ends up in promotion or repression of inflammation [48]. Yet, the mechanism and clinical relevance of MMP-2 suppression and MMP-3 stimulation by tested MAMPs remain unexplained.

In conclusion, we produce evidence to propose that cyclic dinucleotides interact with the early LPS-mediated interleukin and MMP response of gingival fibroblasts, while late response is mainly regulated by LPS. These findings suggest that the action of these microbe-associated molecules may not always be detrimental to the host. Instead, some MAMPs could even be beneficial in maintaining homeostasis between the host and bacteria in the oral cavity via regulating host immune response against pathogens like *P. gingivalis.* We propose that cyclic dinucleotides or their mimics have potential to be used as an adjunctive tool in periodontal therapy.

## 4. Materials and Methods 

### 4.1. Cell Culture

The HGFs, were originally isolated from the extracted wisdom teeth of five young adults (age 18–25 year). Patients were non-smoking, periodontally healthy, and not using any medication with a known effect on the periodontal tissues. All patients gave informed consent before the surgical procedures. The experimental protocol was approved by Ethics committee of the Hospital District of South-West Finland and the Ethical committee of the Dentistry, University of Helsinki (The permission for tissue biopsies was given in 19 November.2002 and the number of the study case was §262) [49]. HGFs were cultured in Dulbecco’s modified eagle medium supplemented with 10% fetal bovine serum (Gibco BRL, Life Technologies), antibiotics (100 IU/mL penicillin and 100 µg/mL streptomycin), and 1% non-essential amino acid (Gibco BRL, Life Technologies), at 37 °C and 5% CO_2_. Culture media were changed three times per week; the cells were passaged when reaching 80–90% confluence.

### 4.2. Synthesis of c-di-GMP and c-di-AMP

The synthesis of cyclic dinucleotides followed the protocol described by Gaffney et al. [50].

### 4.3. Preparation of P. gingivalis LPS Stock Solution

Endotoxin-free water was used to dissolve LPS of *P. gingivalis* (Invivogen, San Diego, CA, USA). Stock concentration was defined as 1 mg/mL.

### 4.4. Incubation of Fibroblasts with Cyclic Dinucleotides

HGFs (3 × 10^5^ /well) were incubated in 12-well plates at 37 °C and 5% CO_2_ until the confluence of the cells reaches 80%. PBS was used to wash the cells three times. The fibroblasts were incubated at 37 °C and 5% CO_2_ for 2 h and 24 h with fresh media containing different concentrations (100 μM, 10 μM, 1 μM) of c-di-GMP or c-di-AMP, either in the presence or absence of *P. gingivalis* LPS (1 μg/mL). The control cells were not incubated with any of the MAMPs. After 2 h and 24 h of incubations, cell culture media were collected and freezed at −70 °C until biochemical analyses. Mechanical scraper and lysis buffer (50 mM Tris-Cl, 150 mM NaCl, and 1% Triton X-100) were used to scrape and lyse the cells. The collected lysates were sonicated for 10 s, and the Bradford method (Bio-Rad, Hercules, CA, USA) was used to determine protein levels of cell lysates.

### 4.5. Analysis of Interleukin and MMP Concentrations

The Luminex technique (Bio-Rad, Santa Rosa, CA, USA) and the commercially optimized Bio-Plex kits (pro-human cytokine group I assay; Bio-Rad, Santa Rosa, CA, USA) were used to detect IL-8, −10, and −34, and MMP-1, −2, and −3 concentrations. All analyses were performed according to the manufacturer’s instructions. The limit of assay detection was 2.7 pg/mL for IL-8, 0.6 pg/mL for IL-10, 51.9 pg/mL for IL-34, 33.7 pg/mL for MMP-1, 39 pg/mL for MMP-2, and 28.5 pg/mL for MMP-3. All data were presented as concentration of each cytokine or MMP per 1 µg of protein. Experiments were done in triplicate. 

### 4.6. Statistical Analysis

IBM SPSS software (version 23, IBM, Armonk, New York, NY, USA) was used in all analyses. In all figures bar charts present values of means and standard deviations. One-way analysis of variance (ANOVA) followed by Bonferroni correction was used to analyze intergroup differences of cytokine and MMP levels. A *p*-value ˂ 0.05 was defined as statistically significant.

## Figures and Tables

**Figure 1 pathogens-09-00792-f001:**
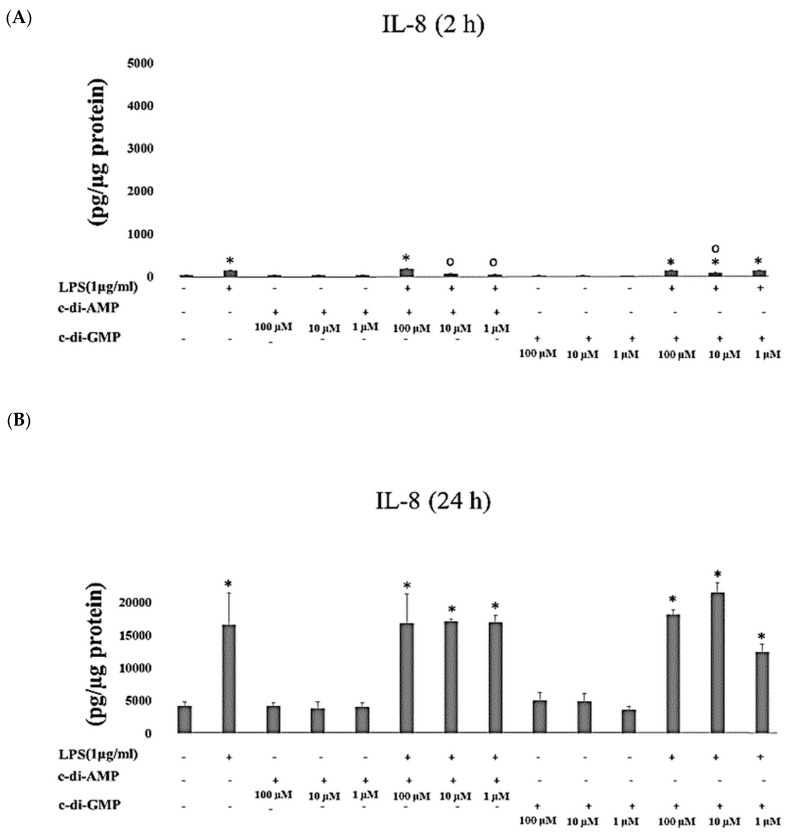
Extracellular levels of IL-8 after incubating Human gingival fibroblasts (HGFs) for 2 h (**A**) and 24 h (**B**) with three test concentrations of c-di-GMP and c-di-AMP either alone or together with *P. gingivalis* LPS. Bars express the mean ± standard deviation for triplicate tests. ∗ indicates a statistical difference with the control (no LPS, c-di-GMP, or c-di-AMP), and **○** indicates a statistical difference with *P. gingivalis* LPS alone.

**Figure 2 pathogens-09-00792-f002:**
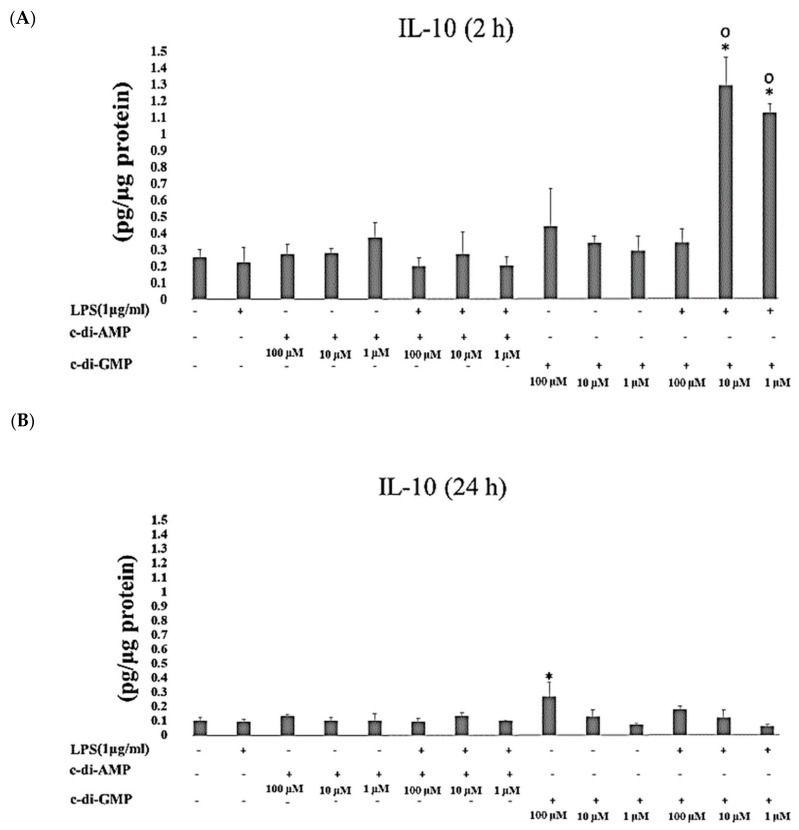
Extracellular levels of IL-10 after incubating Human gingival fibroblasts (HGFs) for 2 h (**A**) and 24 h (**B**) with three test concentrations of c-di-GMP and c-di-AMP either alone or together with *P. gingivalis* LPS. Bars express the mean ± standard deviation for triplicate tests. ∗ indicates a statistical difference with the control (no LPS, c-di-GMP, or c-di-AMP), and **○** indicates a statistical difference with *P. gingivalis* LPS alone.

**Figure 3 pathogens-09-00792-f003:**
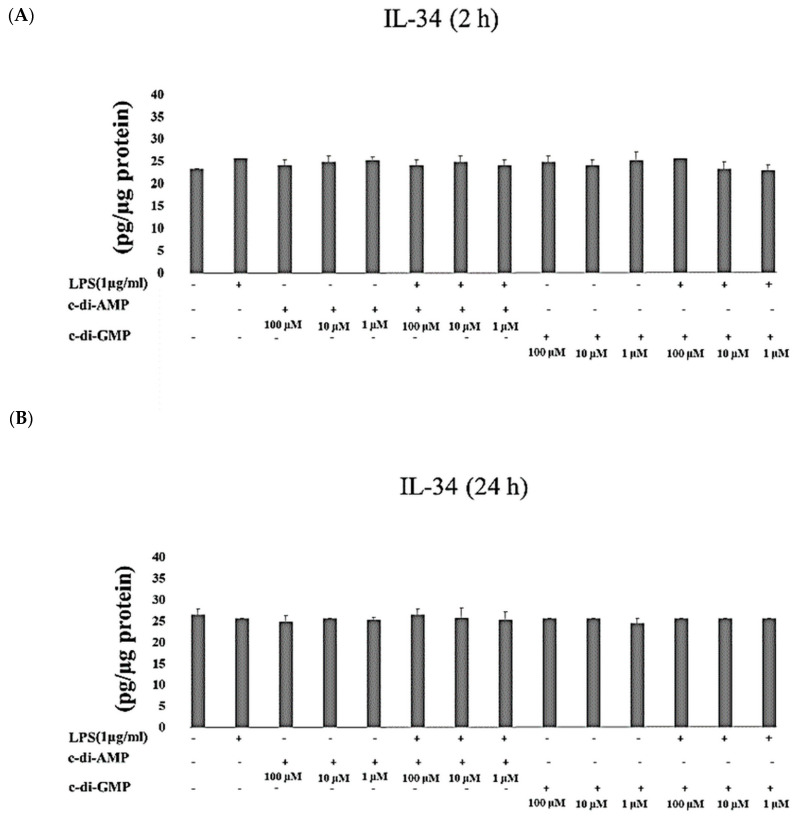
Extracellular levels of IL-34 after incubating HGFs for 2 h (**A**) and 24 h (**B**) with three test concentrations of c-di-GMP and c-di-AMP either alone or together with *P. gingivalis* LPS. Bars express the mean ± standard deviation for triplicate tests.

**Figure 4 pathogens-09-00792-f004:**
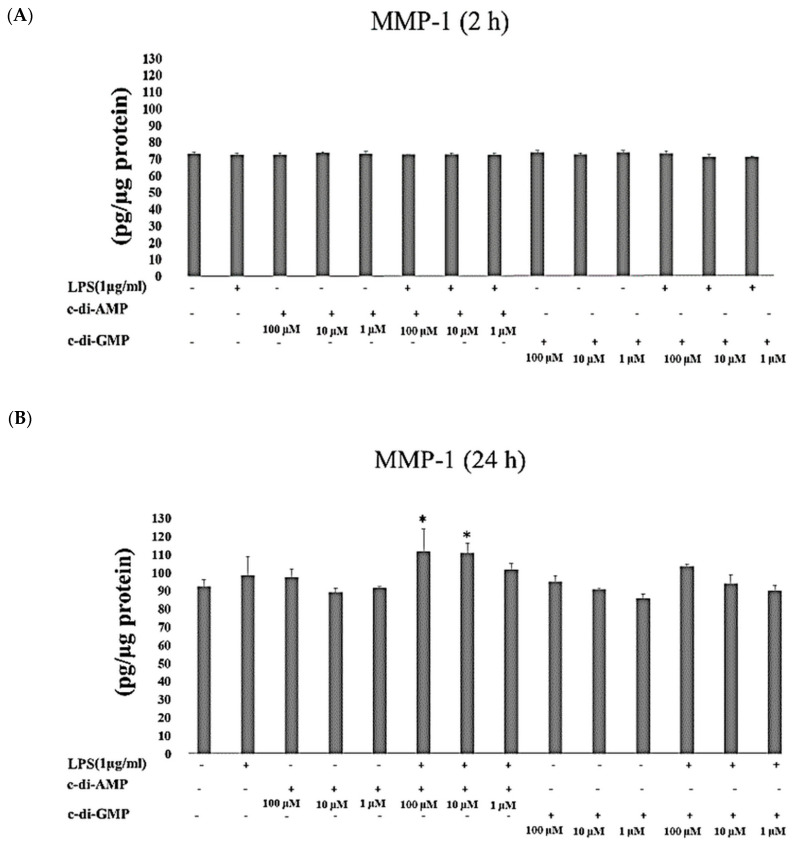
Extracellular levels of MMP-1 after incubating HGFs for 2 h (**A**) and 24 h (**B**) with three test concentrations of c-di-GMP and c-di-AMP either alone or together with *P. gingivalis* LPS. Bars express the mean ± standard deviation for triplicate tests. ∗ indicates a statistical difference with the control (no LPS, c-di-GMP, or c-di-AMP).

**Figure 5 pathogens-09-00792-f005:**
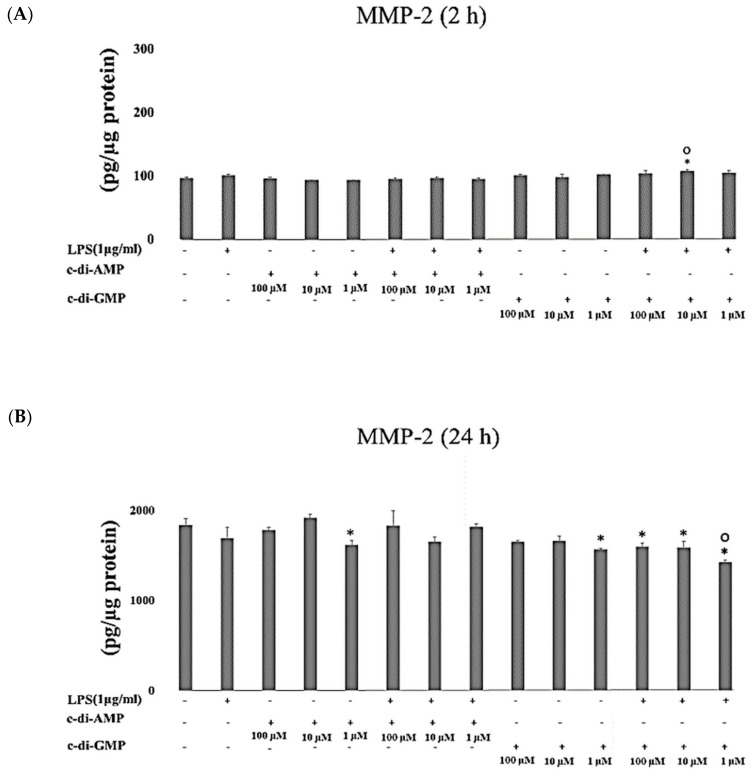
Extracellular levels of MMP-2 after incubating HGFs for 2 h (**A**) and 24 h (**B**) with three test concentrations of c-di-GMP and c-di-AMP either alone or together with *P. gingivalis* LPS. Bars express the mean ± standard deviation for triplicate tests. ∗ indicates a statistical difference with the control (no LPS, c-di-GMP, or c-di-AMP), and **○** indicates a statistical difference with *P. gingivalis* LPS alone.

**Figure 6 pathogens-09-00792-f006:**
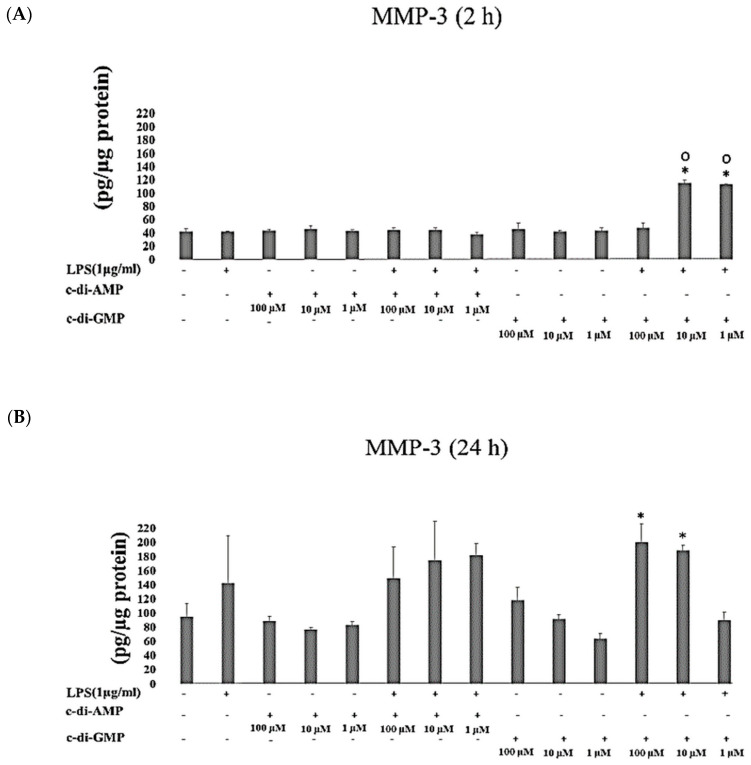
Extracellular levels of MMP-3 after incubating HGFs for 2 h (**A**) and 24 h (**B**) with three test concentrations of c-di-GMP and c-di-AMP either alone or together with *P. gingivalis* LPS. Bars express the mean ± standard deviation for triplicate tests. ∗ indicates a statistical difference with the control (no LPS, c-di-GMP, or c-di-AMP), and **○** indicates a statistical difference with *P. gingivalis* LPS alone.

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
