# Peer review of "Activation of Gingival Fibroblasts by Bacterial Cyclic Dinucleotides and Lipopolysaccharide"

_pathogens, 2020, doi:10.3390/pathogens9100792_

Round 1
Reviewer 1 Report
Thank you for this nice artikel.
But I think that the introduction is to general, what is the main topic of the study? Is it possible to make it more focused?
And what is the relevance of in the long run? Clinical use? What is the purpose of this study, please clarify.
Why the time points 2 and 24h? Has any other studies used the same time points and definitions (early and late)? Referentens please.
The limitations of this study are quite many and significant. Is it still reasonable draw these conclusions? Please elaborate.
Author Response
The authors have revised the manuscript” Activation of Gingival Fibroblasts by Bacterial Cyclic Dinucleotides and Lipopolysaccharide" (pathogens-896162), by Samira Elmanfi et al. according to the reviewers’ comments and suggestions. A response to each of their comments is presented below (underlined). The changes in the revised manuscript are in colored front. We are grateful for their useful remarks that have improved the quality of our manuscript.
Reviewer: 1
I think that the introduction is to general, what is the main topic of the study? Is it possible to make it more focused?
- Introduction section is modified to bring the main topic of study front (please see introduction).
What is the relevance of in the long run? Clinical use? What is the purpose of this study, please clarify.
- Clinical relevance is included into the text (page 10, line 358-362).
Why the time points 2 and 24h? Has any other studies used the same time points and definitions (early and late)? References please.
- Selection criteria of the time points are explained in the text (page 9, line 279-286).
The limitations of this study are quite many and significant. Is it still reasonable draw these conclusions? Please elaborate.
- Conclusion is re-written (page 10, line 356-358).
Reviewer 2 Report
Manuscript Review Comments
Title: “Activation of Gingival Fibroblasts by Bacterial Cyclic Dinucleotides and Lipopolysaccharide” (pathogens-896162)
The authors present a study in which they perform an assessment of the effect of cyclic dinucleotides and P. gingivalis LPS on gingival fibroblasts. Although it is an interesting topic, and this reviewer has found several shortcomings in the manuscript that should be addressed.
Comment to the authors
The manuscript presents a serious overlap percentage (25%) with other one previously published by the same authors: Elmanfi S, Zhou J, Sintim HO, Könönen E, Gürsoy M, Gürsoy UK. Regulation of gingival epithelial cytokine response by bacterial cyclic dinucleotides. J Oral Microbiol. 2019;11(1):1538927. After reading both manuscripts I understand that they address different but related topics. But since the overlapping includes results section and other parts of the manuscript, I would recommend the author to make an extensive rewriting of the text to avoid potential plagiarism issues, prior to any deeper evaluation.
Also, the study does not provide any ethical committee approval, although the sample for the study was obtained from a patient. I think this is mandatory even in this kind of studies.
Author Response
The authors have revised the manuscript” Activation of Gingival Fibroblasts by Bacterial Cyclic Dinucleotides and Lipopolysaccharide" (pathogens-896162), by Samira Elmanfi et al. according to the reviewers’ comments and suggestions. A response to each of their comments is presented below (underlined). The changes in the revised manuscript are in colored front. We are grateful for their useful remarks that have improved the quality of our manuscript.
Reviewer: 2
The manuscript presents a serious overlap percentage (25%) with other one previously published by the same authors: Elmanfi S, Zhou J, Sintim HO, Könönen E, Gürsoy M, Gürsoy UK. Regulation of gingival epithelial cytokine response by bacterial cyclic dinucleotides. J Oral Microbiol. 2019;11(1):1538927. After reading both manuscripts I understand that they address different but related topics. But since the overlapping includes results section and other parts of the manuscript, I would recommend the author to make an extensive rewriting of the text to avoid potential plagiarism issues, prior to any deeper evaluation.
- To avoid potential self-plagiarism, significant changes were done on the text. All changes are yellow highlighted.
The study does not provide any ethical committee approval, although the sample for the study was obtained from a patient. I think this is mandatory even in this kind of studies.
- Ethical committee approval information added to the manuscript (page 11, line 371-374).
Reviewer 3 Report
Dear author,
Although it is an in vitro study, congratulations for the work.
I advice:
Line 410: ”… data; the amount…” Remove ;
Materials and Methods
Line 382: Fibroblasts were removed from the gums of extracted teeth. What teeth are you referring?
Have patients given informed consent to use the gingival tissue?
How many patients were used? How old are the patients?
Were inclusion/exclusion criteria used for the selection of patient?
Best regards,
Author Response
The authors have revised the manuscript” Activation of Gingival Fibroblasts by Bacterial Cyclic Dinucleotides and Lipopolysaccharide" (pathogens-896162), by Samira Elmanfi et al. according to the reviewers’ comments and suggestions. A response to each of their comments is presented below (underlined). The changes in the revised manuscript are in colored front. We are grateful for their useful remarks that have improved the quality of our manuscript.
Reviewer: 3
Line 410: ”… data; the amount…” Remove;
- Removed (page 11, line 403).
Materials and Methods
Line 382: Fibroblasts were removed from the gums of extracted teeth. What teeth are you referring?
- Information regarding the tissue sampling is included into the text (page 11, line 368).
Have patients given informed consent to use the gingival tissue?
- Yes, information is included into the text (page 11, line 370, 371).
How many patients were used? How old are the patients? Were inclusion/exclusion criteria used for the selection of patient?
- Patients were five young adults with an age range of age 18-25 years This information is included into the text (page 11, line 368, 369).
- Inclusion and exclusion criteria are included into the text (page 11, line 369,370).
Round 2
Reviewer 1 Report
I think that the introduction can be more focused. Other than that well done.
Reviewer 2 Report
Authors have correcly addressed my previous comments and the ones from my fellow reviewers.